# The Unnoticed Threat: Clinical Characteristics, Risk Factors, and Outcome of Mucormycosis in Solid Organ Transplantation

**DOI:** 10.3390/jof11120853

**Published:** 2025-11-29

**Authors:** Jorge Boán, Mario Fernández-Ruiz, Eduardo Aparicio-Minguijón, José María Aguado

**Affiliations:** 1Unit of Infectious Diseases, “12 de Octubre” University Hospital, Instituto de Investigación Sanitaria Hospital “12 de Octubre” (imas12), 28041 Madrid, Spain; mario_fdezruiz@yahoo.es (M.F.-R.); edu.ap93@gmail.com (E.A.-M.);; 2Centro de Investigación Biomédica en Red de Enfermedades Infecciosas (CIBERINFEC), Instituto de Salud Carlos III (ISCIII), 28037 Madrid, Spain; 3School of Medicine, Universidad Complutense, 28040 Madrid, Spain

**Keywords:** mucormycosis, solid organ transplantation, epidemiology, risk factors, treatment, outcome

## Abstract

Mucormycosis is an uncommon but life-threatening invasive fungal infection caused by molds of the order Mucorales, whose incidence has increased among solid organ transplant (SOT) recipients in recent years. Profound immunosuppression (particularly high-dose corticosteroids), T-cell-depleting therapies, diabetes mellitus, and previous episodes of graft rejection are the main predisposing conditions. This narrative review summarizes the current evidence on epidemiology, pathogenesis, risk factors, clinical presentation, diagnostic strategies, and treatment outcomes of mucormycosis in the SOT population. Pulmonary and rhino-orbital-cerebral infections are the predominant clinical forms, often characterized by rapid angioinvasive progression and mortality rates exceeding 45%. Early diagnosis remains challenging due to nonspecific clinical manifestations and the limited sensitivity of conventional diagnostic tools, although molecular techniques such as the detection of circulating Mucorales DNA in blood and metagenomic next-generation sequencing are promising. Liposomal amphotericin B remains the first-line therapy, ideally associated to surgical debridement and reduction in immunosuppression, while broad-spectrum triazoles (isavuconazole and posaconazole) represent alternative or salvage options. Despite recent advances in diagnostic methods and antifungal therapy, the prognosis of post-transplant mucormycosis remains poor, underscoring the need for multidisciplinary management and collaborative studies to inform the clinical management in this high-risk population.

## 1. Introduction

Mucormycosis refers to a range of opportunistic, aggressive, invasive fungal infections (IFIs) caused by filamentous fungi belonging to the subphylum Mucoromycotina, order Mucorales. The etiological agents of mucormycosis are ubiquitous fungi commonly found in the environment, such as decaying organic matter [1]. Mucorales characteristically produce broad, sparsely septate, ribbon-like hyphae with an irregular diameter of 5–20 μm. Spores are easily aerosolized and dispersed, causing infection in humans when inhaled or through cutaneous or percutaneous route [1]. In healthy individuals, innate immunity alone is sufficient to prevent infection, except in cases of massive contamination with contaminated soil and soft-tissue inoculation due to high-energy trauma. During infection, the hyphae invade blood vessels, leading to infarction, necrosis, and thrombosis [2].

In recent years, mucormycosis has been established as an important form of IFI in solid organ transplant (SOT) recipients, in whom it often shows an aggressive clinical course and substantial mortality rates [2,3,4]. Although SOT is typically recognized as a risk factor [5,6], the diagnosis of post-transplant mucormycosis is often missed until an advanced stage of disease due to the lack of awareness by transplant physicians, which is reinforced by the low incidence and the poor characterization of SOT recipients at increased risk of this life-threatening complication. The present review is aimed at providing a comprehensive, updated summary of the epidemiology, pathogenesis, clinical presentation, diagnostic approach, and treatment of mucormycosis in the SOT population. A series of PubMed searches were conducted by September 2025 using the medical subject headings (MeSH) “solid organ transplantation” (and associated terms) AND “mucormycosis” OR “zygomycosis” OR “mucorales”, either alone or combined with other MeSH terms and restricted by publication date to the past 20 years. We also reviewed the most recent global guidelines on mucormycosis [7], as well as guidelines focused on other high-risk groups such as hematological patients and hematopoietic stem-cell transplant (HSCT) recipients [8,9].

## 2. Epidemiology

The epidemiology of IFI due to filamentous fungi in immunocompromised patients has evolved with changing practices in antifungal prophylaxis and diagnostic mycology laboratories [10]. Mucorales are, following *Aspergillus* spp., the most common pathogenic molds in the SOT population [11,12]. Although only a few genera within this order have been conventionally reported as predominant in clinical cases (e.g., *Rhizopus*, *Lichtheimia*, or *Mucor*), culture recovery from infected tissue samples is suboptimal and may have skewed our current understanding of the etiologic spectrum of mucormycosis (Table 1). In fact, the number of species involved in cases of human mucormycosis has expanded considerably over the past decades, in parallel with improvements in culture-based morphological identification and the widespread application of more accurate molecular methods. Notwithstanding this, the incidence of proven or probable mucormycosis in multicenter cohorts and large single-center series has maintained overall as low (from 0.1% to 2.5%) [13,14,15,16,17,18,19,20] during the past two decades (Table 2).

## 3. Pathogenesis

The primary mode of acquisition of mucormycosis is through the inhalation of spores from environmental sources [22]. Most Mucorales sporangiospores are small enough (3–11 μm) to evade the upper respiratory tract mechanical defenses and reach the distal alveolar spaces [1]. Larger spores (>10 μm) may deposit in the nasal turbinates, causing sinusitis [22]. Inhalation of a high spore inoculum, which typically occurs during excavation or construction activities or while working on contaminated air ducts, can result in progressive forms of pulmonary mucormycosis even in immunocompetent hosts [23] (Figure 1).

Little is known about the interactions between Mucorales and epithelial cells, the first step of the process through which sporangiospores attach to the respiratory and gastrointestinal tracts [24]. It has been shown that *Rhizopus* spores can adhere to extracellular matrix proteins such as laminin and type IV collagen [25] that embeds epithelial or endothelial cells. In vitro studies have demonstrated that Mucorales can induce host cell damage due to the presence of toxins, which would account for the distinctive clinical feature of extensive tissue necrosis [24,26]. Mucorales also possess a mechanism for adhering to and invading endothelial cells through the specific recognition of the heat shock glucose-regulated protein 78 (GRP78), whose expression on the endothelial cell surface is enhanced with elevated available serum iron and glucose levels. Mucorales germlings, but not spores, are able bind to GRP78, initiating endothelial-cell-mediated fungal endocytosis and ultimately leading to host cell death [27]. The spore coat protein (CotH)—which is universally present in Mucorales and absent from other pathogens [28]—is the key GRP78 ligand on the host endothelium and plays a pivotal role in angio-invasion [29]. Endothelial cell uptake is also enhanced upon the activation of the platelet-derived growth factor (PDGF) signaling system [28].

Mucorales exhibit a remarkable angioinvasive capacity that leads to vessel wall necrosis and mycotic thrombi, with the subsequent occurrence of infarction and hematogenous dissemination [30]. Infected tissue typically reveals extensive necrosis with diffuse polymorphonuclear leukocyte infiltration, although inflammation may be minimal in ischemic necrotic areas despite the presence of numerous hyphae [31,32,33,34].

In order to establish angioinvasive infection, Mucorales spores must overcome phagocytosis by macrophages and neutrophils to germinate into hyphae [35]. Therefore, mucormycosis rarely affects individuals with a preserved innate immune response, but it can be rapidly progressive for neutropenic patients or those with impaired macrophage function or on chronic corticosteroid therapy [36,37,38]. Both mononuclear and polymorphonuclear phagocytes prevent the germination of Mucorales spores, whereas natural killer (NK) cells induce perforin-mediated damage of the hyphal forms [39,40]. Neutrophil exposure to *R. oryzae* hyphae triggers a robust proinflammatory gene expression through the upregulation of Toll-like receptor 2 and the induction of nuclear factor kappa B (NF-κB) [41]. Hyphal damage is mediated through oxidative mechanisms upon monocyte or neutrophil attachment [42]. Platelets may also contribute to host defense by adhering to hyphae and secreting microbiocidal peptides [43].

Similarly to other susceptible patient groups, the ability of the innate immunity to inhibit spore germination is compromised in SOT recipients, which allows fungal growth due to decreased phagocytic cell numbers (i.e., neutropenia induced by immunosuppressive or antiviral agents) or function [24]. In the setting of this impaired host response, Mucorales can rapidly spread through tissues and blood vessels [24,35]. Long-term use of corticosteroids constitutes a well-recognized risk factor for post-transplant mucormycosis [44] through the complex quantitative and qualitative immunosuppressive effects exerted by this therapy and the associated hyperglycemic state [38].

There is limited evidence for a major role of the adaptive immune system in the pathogenesis of mucormycosis, since human immunodeficiency virus (HIV) infection alone is not a predisposing condition and T-lymphocyte depletion does not increase susceptibility to Mucorales in mice [45,46]. Nevertheless, mucorales-specific CD4+ or CD8+ T-cells producing interferon (IFN)-γ, interleukin (IL)-10, and, to a lesser extent, IL-17 can be detected in response to hyphae and during invasive mucormycosis [47,48,49]. Mucorales have also been reported to activate human dendritic cells through the fungal pattern recognition receptor Dectin-1, resulting in IL-23 production and induction of proinflammatory Th17 responses [50].

Recent evidence suggests that Mucorales, as intracellular pathogens, can persist within granulomatous clusters, supporting the notion of a latent infection that would reactivate during periods of intensive immunosuppression such as SOT [24]. This possibility raises the question of whether SOT candidates or recipients should be screened for latent mucormycosis in order to initiate preemptive antifungal therapy [47].

## 4. Risk Factors

Underlying conditions and risk factors classically associated with mucormycosis include poorly controlled diabetes mellitus, chronic corticosteroid therapy, hematological malignancies with severe neutropenia (absolute neutrophil count <500 cells/μL), malnutrition, burns, high-energy penetrating trauma, and iron overload [51]. Coronavirus disease 2019 (COVID-19) has recently emerged as a predisposing condition, particularly in diabetic patients with prolonged intensive care unit (ICU) admission, with geographical predominance of the Indian subcontinent [52,53].

The overall incidence of mucormycosis after SOT is much lower than that observed in the HSCT population [12,54,55] and, compared to diabetic and onco-hematological patients, SOT recipients comprise only a small proportion in large multicenter cohorts [6]. The majority of early cases occurring within the first 100 post-transplant days are associated with enhanced immunosuppression for the treatment of acute graft rejection [56]. Some old studies reported a temporal relationship between the use as prophylaxis of antifungal agents with no in vitro activity against Mucorales, such as voriconazole (VORI), and the occurrence of mucormycosis [57,58], although the evidence on a firm causal association remains inconclusive. In a recent systematic review of cases of mucormycosis in SOT recipients published between 2002 and 2022 (n = 183), the most frequent complication reported to precede the diagnosis was graft rejection (30.9%), while one-fourth of patients had previously undergone reoperation or re-transplantation. Regarding the type of immunosuppressive therapy, 71.6% of patients were receiving more than three agents, with predominance of corticosteroids, tacrolimus, and mycophenolic acid. One-third had received antifungal prophylaxis, which was potentially effective against Mucorales (amphotericin B [AmB]) in 22.6% of cases. Likely reflecting the relative proportion of SOT procedures performed worldwide, kidney transplantation (KT) accounted for half of the cases (50.8%), followed by heart (HT) and liver transplantation (LT), with 18.6% and 16.9% of patients, respectively [21].

In a registry-based report from the US Renal Data System (USRDS) that included 306,482 KT recipients between 1988 and 2015, 222 (0.07%) were diagnosed with proven mucormycosis. Age ≥ 65 years, Hispanic ethnicity, deceased donation, and tacrolimus-containing immunosuppressive regimen were identified as risk factors, whereas the use of mycophenolate mofetil or azathioprine was found to be protective, as well as (quite surprisingly) the presence of iron overload [59]. This study, however, is limited by large study period, which spanned three decades, and the lack of data granularity.

In a single-center cohort of 603 lung transplant (LuT) recipients, 26.4% of patients suffered 182 episodes of invasive fungal infection (IFI), including 4 cases of mucormycosis (2.2%). The use of mold-active prophylaxis for patients with chronic airway fungal colonization before transplantation was associated with lower risk of early (≤90 days) but not late invasive mold infection (IMI), whereas renal replacement therapy increased the risk of both early and late forms [60]. In a systematic literature review specifically focused on LuT recipients, additional risk factors included diabetes mellitus (present in 38% of 121 cases of post-transplant mucormycosis), renal failure (28%), treatment with high-dose corticosteroids (15%), diagnosis within the previous 6 months of acute graft rejection (8%), diagnosis of post-transplant lymphoproliferative disorder (10%), and previous therapy with rituximab (6%) [61].

The literature regarding specific risk factors for mucormycosis in the setting of LT is scarce and includes diabetes mellitus (particularly in the form of ketoacidosis), renal failure [62,63], cholestasis, high blood transfusion requirements [64], acute rejection [62,65], treatment with high dose of steroids or anti-CD3 antibody (OKT3) [62], ABO-incompatible transplantation [66], bacterial infection [63,67,68], and re-transplantation [65]. In a small single-center retrospective study that included 51 LT recipients, the authors suggested that the poor clinical condition of patients before transplantation (with prolonged hospitalization and decompensated cirrhosis) and the break in aseptic techniques during organ procurement may have contributed to the exceptionally high incidence of mucormycosis reported (7.8% over a 10-year period) [69]. Of note, fatal cases of gastrointestinal mucormycosis have been reported in LT recipients from living donors due to acute liver failure or suffering early graft dysfunction [70].

We are not aware of previous studies that have investigated the predisposing conditions for other SOT groups, such as HT or multivisceral transplantation.

## 5. Clinical Presentation

The clinical spectrum of post-transplant mucormycosis is wide, depending on the immune status and comorbidities of the susceptible host and the type of graft, although symptoms and signs are often nonspecific. Mucormycosis can present after SOT as a fulminant angioinvasive disease with frequent dissemination and fatal outcomes. The most common forms in this patient group are rhino-sinusal, pulmonary, rhino-orbital-cerebral, cutaneous, and disseminated disease [71,72,73]. In the recent literature review by Palomba et al., the predominant clinical manifestations were pulmonary and rhino-orbital-cerebral (ROC) mucormycosis, which accounted for 24.6% of cases each, whereas 10% of patients had disseminated infection [21]. In the meta-analysis by Jeong et al. [6], which covered the clinical manifestations of mucormycosis across different patient populations, SOT was associated with an increased risk of pulmonary, gastrointestinal, or disseminated infection in comparison with ROC mucormycosis. The higher proportion of gastrointestinal involvement—17% in SOT recipients [21] versus 8% in the comprehensive review of published cases [6]—may be explained by the impact of intraabdominal procedures in the setting of LT and multivisceral transplantation. In the same line, the lower respiratory tract was the most common site of disease in LuT recipients (62%), followed by disseminated (13%) and ROC infection (12%) [62]. Regarding the timing after transplantation, two thirds of the cases occur within the first 100 days (the period of highest immunosuppression), whereas one-fifth is diagnosed beyond month 12 [21].

### 5.1. Pulmonary Mucormycosis

Clinical manifestations may be indistinguishable from those observed in more common forms of post-transplant IMI, such as invasive pulmonary aspergillosis, a pneumonia that does not resolve despite broad-spectrum antibiotics with refractory fever, nonproductive cough, progressive dyspnea, and pleuritic chest pain [35,47,74,75,76,77]. Pulmonary mucormycosis may invade through pulmonary tissue planes to involve the bronchi, diaphragm, chest wall, and pleura [76]. Hyphal invasion of blood vessels results in necrosis of the surrounding parenchyma, ultimately leading to cavitation or potentially fatal hemoptysis [22,78,79,80,81]. Atypical presentations include mycotic pulmonary artery aneurysms and pseudoaneurysms, bronchial obstruction, or asymptomatic solitary nodules [22,74].

A history of antibiotic-refractory pneumonia in an SOT recipient with previous exposure to antifungal agents with no activity against Mucorales, suggestive findings in the thoracic computed tomography (CT) scan (such as a reverse halo sign and multiple nodular infiltrates with or without pleural effusion), and the repeated absence of detectable *Aspergillus* galactomannan (GM) antigen in serum or bronchoalveolar lavage fluid (BALF) samples should raise the suspicion of pulmonary mucormycosis [82,83,84,85]. Radiological features are variable, including focal consolidations, cavitary lesions, and rapidly evolving diffuse opacities [85,86]. In the systematic review by Palomba et al., lobar consolidations and cavitary lesions were the most common findings in chest imaging (35% and 29%, respectively), followed by solitary pulmonary nodules [21].

### 5.2. Rhino-Orbital-Cerebral Mucormycosis

Rhino-sinusal, rhino-orbital, and ROC mucormycosis are clinical forms typically associated with uncontrolled diabetes (in particular, ketoacidosis) and profound neutropenia, although they can also appear in the SOT setting [5,6,21,35,82,87,88]. Following inhalation of spores, the infection is initially confined to the nasal turbinates and paranasal sinuses to rapidly progress to the orbital apex and roof, with the appearance of characteristic necrotic eschars in the nasal cavity, the palate, or even the face as early diagnostic signs [73,81,89,90,91]. It should be noted, however, that necrotic nasal or palate lesions are clinically evident within the first days from the onset of infection in only 50% of patients [92]. Sinus mucormycosis may spread to adjacent structures. Maxillary and ethmoid infection often extend into the orbit, whereas the cavernous sinus, temporal lobe, or carotid artery are involved in cases initiated in the sphenoid sinus. Orbital invasion may lead to unilateral swelling, visual loss, and cranial nerve palsies. Intracranial complications include abscesses, sinus thrombosis, and, more rarely, meningitis [35,93,94,95,96,97]. As compared to diabetic patients with ROC mucormycosis [98], SOT recipients had a lower likelihood of orbital (80.0% versus 34.8%) and sinonasal (100% versus 89.9%) involvement but a higher likelihood of central nervous system (CNS) invasion (46.1% vs. 31.4%) [73]. Maxillary and ethmoid sinuses are the most common locations, as well as frontal and temporal lobes in cases of CNS infection [73].

Imaging findings are often suggestive but lack specificity. Sinus CT scan may typically reveal mucosal thickening, air-fluid levels, or bony erosion [99]. The presence of pansinusitis in an immunocompromised patient is highly suggestive of IMI [75,100]. Although soft tissue involvement may be observed in CT scan, magnetic resonance imaging (MRI) provides better resolution to define orbital and intracranial lesions, cavernous sinus thrombosis, and thrombosis of the cavernous portions of the internal carotid artery [81,101].

### 5.3. Cutaneous and Soft Tissue Mucormycosis

Cutaneous mucormycosis can occur as primary infection—following local inoculation at the sites of surgical incision or drain, skin trauma, or intravenous catheter insertion—or secondary to hematogenous dissemination [102]. The use of adhesive polyethylene tape has been reported as a cause of outbreaks in hematological patients [103]. Lesions may present as a black necrotic eschar with surrounding cellulitis, thrombophlebitis, or extension to deep structures (which complicates up to 44% of cases of primary cutaneous involvement) [5]. A literature review [102] identified cases reported in KT, LT, LuT, and HT recipients [54,104,105,106,107,108,109,110], with the infection primarily located at the site of surgical incision in most patients [44,111,112]. This form appears to occur early after transplantation (median interval of 23 days) [102].

## 6. Diagnostic Approaches

The diagnosis of mucormycosis is challenging since clinical manifestations are often nonspecific. In addition, the isolation of Mucorales from clinical samples may reflect culture contamination due to the ubiquitous environmental nature of the fungus. Therefore, histological documentation of tissue invasion has been required for definitive diagnosis [22,113]. The hyphae of the Mucorales are broad (5–20 μm), thin-walled, non-septate, or pauci-septate, with non-dichotomous irregular branching occasionally at right angles (Figure 2) [22,74,114]. Conventional diagnostic methods, such as fungal culture and histopathology, remain the gold standard but are often limited by their slow turnaround times and low sensitivity [115]. These drawbacks are also applicable to the specific SOT population.

### 6.1. Imaging

No radiological sign is sensitive and specific enough to effectively rule in or out pulmonary mucormycosis [7]. The finding in the thoracic CT scan classically associated with this condition is the so-called “reversed halo sign”, an area of central ground-glass necrosis, surrounded by a ring of consolidation reflecting central lung infarction with dense peripheral hemorrhage (Figure 3) [116,117,118]. The presence of multiple (≥10) nodular infiltrates, pleural effusion, concomitant sinus disease, or vessel occlusion in a patient with negative serum and BALF GM assay should raise the suspicion of pulmonary mucormycosis, although this vignette has been mainly characterized in neutropenic patients rather than in SOT recipients [76,119,120,121].

Follow-up imaging has been used to assess therapeutic response, as the decrease in ground-glass opacities surrounding a reversed halo, central necrotic cavity, or air-crescent sign may occur with recovery (Figure 4). Nevertheless, the absence of radiographic changes during the first 30 days of therapy is not necessarily predictive of poor outcome [77,86,122,123].

The most common finding in patients with ROC mucormycosis is sinusitis, which may not be distinguishable from bacterial infection. While mucosal thickening and partial or complete sinus opacification are frequent (Figure 5), bony erosion constitutes an uncommon and late finding. MRI is more sensitive than CT scan to detect orbital and brain involvement with edema in the orbital muscles, the most common finding of orbital disease [124,125,126,127]. Notably, Mucorales can spread through the lamina papyracea to involve the orbit or the ethmoid sinus to reach the cavernous sinus without apparent bony destruction [128].

### 6.2. Antigen Biomarkers

Most Mucorales produce low amounts of (1 → 3) ß-D-glucan (BDG), below the limit of detection of commercially available assays, although some species of *Rhizopus* can yield positive results [129,130,131,132,133,134]. Therefore, BDG testing is not recommended as a diagnostic tool for mucormycosis [7,135]. On the other hand, GM levels are usually low since Mucorales do not expose cell wall polysaccharides on the hyphae surface [136]. A previous study reported an unexpectedly high rate of serum or BALF GM positivity in patients with gastrointestinal mucormycosis, although co-infection with *Aspergillus* spp. or false-positive results could not be ruled out [137]. Novel diagnostic tools, such as detection of serum dihexasaccharide [138], a pan-fungal monoclonal antibody (2DA6) recognizing the α-1,6-linked mannose [139] or the enumeration of Mucorales-specific T-cells [48,140], are still in the phase of clinical research.

### 6.3. Conventional Diagnostic Methods

Direct microscopical examination demonstrating the presence of fungal elements is sufficient to establish the diagnosis of IFI. Therefore, this approach is recommended for all tissues and fluids from normally sterile sites. Direct microscopy for fungi should be available within 2–4 h from arrival at the Microbiology laboratory to ensure early diagnostic testing [141]. Optical brighteners (e.g., Calcofluor–White) should be considered for rapid direct microscopy (Figure 2) given their improved sensitivity and broad availability [7,142]. Fungal culture remains the foundation for the laboratory diagnosis, allowing the isolation, identification, and antifungal susceptibility testing of the pathogenic mold [142].

### 6.4. Molecular-Based Methods for Direct Detection

Polymerase chain reaction (PCR) assays detect fungal DNA/RNA with high sensitivity and specificity, making it a valuable tool for the diagnosis of IFI [143]. An increasing number of studies have shown that molecular detection of Mucorales DNA in tissue samples has clinical utility in both fresh [144,145,146,147,148,149,150,151,152,153,154,155] and formalin-fixed paraffin-embedded specimens [2,142,146,148,151,156,157,158,159,160,161,162]. Various Mucorales PCR assays are commercially available for use in different specimens (such as serum, respiratory samples, or tissue) and offer a shorter turnaround time than pan-fungal 18s rDNA-directed PCRs [56].

Significant progresses have been made in recent years for the non-invasive diagnosis of mucormycosis based on the amplification of circulating Mucorales DNA in serum samples. The MODIMUCOR study was a prospective multicenter study carried out in France that enrolled 232 patients with suspicion of IMI. A quantitative PCR (qPCR) detecting *Lichtheimia* spp., *Rhizomucor* spp., and both *Rhizopus* spp. and *Mucor* spp. (without distinguishing between these two genera) performed on serum samples collected twice a week enabled for the early diagnosis of mucormycosis in 40 patients (15% of which were SOT recipients). Interestingly, qPCR positivity predated for a median of four days the sampling of the first mycological or histological positive specimen. In addition, a negative qPCR within the first week after the initiation of AmB-based therapy was associated with higher patient survival [163]. In a recent systemic review and meta-analysis of 30 studies, Mucorales PCR in BALF yielded a specificity of 97.5% and specificity of 95.8%. Although the sensitivity decreased to 81.6% when blood specimens (i.e., whole blood, plasma, and serum) were used, the specificity (95.5%) and positive likelihood ratio (18.3) remained high [164]. The performance of this approach may be compromised due to the low levels of circulating DNA in blood, although the sensitivity of most assays improves when larger serum volumes are analyzed [165]. On the other hand, Mucorales PCR testing has not yet been included as a mycological diagnostic criterion in the consensus definitions of IMI [166].

Since only a limited number of diagnostic tools are available and molecular testing remains limited to conventional PCR, metagenomic next-generation sequencing (mNGS) in peripheral blood has been investigated as an innovative non-invasive method [167,168]. In a small retrospective study, the detection of Mucorales sequences by peripheral blood mNGS exhibited an acceptable positive predictive value (72.6%) and a short turnaround (24–48 h), although a relatively high rate of false-positive results occurred, likely due to contamination during sampling or sample processing [167]. It should be noted that SOT recipients were underrepresented in the few studies that have evaluated to date the role of mNGS for the diagnosis of IMI [169,170,171].

## 7. Treatment

Successful treatment of mucormycosis relies on timely diagnosis, reversal of underlying predisposing factors, early surgical debridement of infected tissue, and rapid initiation of effective high-dose systemic antifungal therapy [172]. Global guidelines on mucormycosis [7] remarks the pivotal role of early surgical debridement with clean margins and the immediate initiation of antifungal therapy with liposomal AmB (L-AmB) at a dose of 5–10 mg/Kg per day. In case of CNS involvement, L-AmB should be started at an initial dose of 10 mg/Kg, avoiding slow progressive escalation. Mold-active triazoles (isavuconazole [ISA] or posaconazole [POS]) are favored over polyenes for patients with preexisting renal failure [7,72,173,174,175].

There are few large series that have evaluated the therapy of mucormycosis in the specific population of SOT recipients, in contrast to the HSCT setting [8]. Since both surgical and medical treatments are simultaneously or sequentially performed, it is difficult to ascertain the relative efficacy of antiviral therapy alone [22]. Sun et al. analyzed 90 SOT recipients with ROC mucormycosis from a single center and performed a literature review up to 2009 to conclude that treatment with a lipid AmB formulation in addition to surgical debridement was associated with an improved survival as compared to the deoxycholate form [73]. Another literature review that included 174 cases of mucormycosis in KT recipients published up to 2015 confirmed the benefit provided by lipid formulations, resulting in a survival rate of 73.4% (versus 47.4% with AmB deoxycholate) [176]. In a series of 31 SOT recipients (mainly KT and LT) with pulmonary mucormycosis, 45.2% of the patients received either L-AmB or AmB lipid complex (ABLC) and exhibited an all-cause 90-day mortality rate that was relatively low (16.7%), although the lack of multivariable adjustment due to the low number of patients did not allow the effect of potential confounders to be ascertained, such as disseminated disease or surgical debridement [72].

Even with the lipid formulations, the development of AmB-related nephrotoxicity constitutes a threat for KT recipients and other vulnerable populations [177]. Forrest et al. reported graft and patient outcomes in six KT recipients that were treated with ABLC at doses of 5–10 mg/Kg per day. The only three surviving patients received a median duration of therapy of 89 days and reduction in immunosuppression, leading to graft loss and permanent return to dialysis in all of them [175]. In one-third of the cases reported by Sun et al. the diagnosis of mucormycosis was followed by acute rejection, highlighting the risk of alloreactivity derived from the reduction or discontinuation of immunosuppression [72]. The choice of doses of L-AmB higher than 5 mg/Kg per day should take into account that a sizable proportion of patients will develop renal injury requiring subsequent dose reduction or switching to triazole therapy within the first weeks of therapy [47].

Regarding broad-spectrum triazoles, data on the effectiveness and safety of POS and ISA is limited, since no comparative studies have been conducted and separate outcomes are not usually reported for first-line and salvage uses [173]. A complete or partial response was observed in the four SOT recipients included in two open-label compassionate trials that evaluated POS (as oral suspension) as salvage therapy [178]. A literature review focused on LuT found a clinical response rate of 85.7% for the 14 patients with mucormycosis treated with posaconazole (either alone or associated with AmB-based formulations) [61]. A similar survival rate (92.3%) was observed in a literature review with 13 KT recipients [176]. Delayed-release gastro-resistant tablets and the intravenous formulation of POS are preferred over oral suspension for most patients due to their more predictable pharmacokinetics properties [179].

ISA was approved by US and European regulatory agencies in 2015 for the treatment of mucormycosis based on an open observational study with 21 patients (only one SOT recipient) that received this agent as primary therapy for proven or probable disease [180]. Since then, growing but still limited experience has been gained for this indication in the SOT setting [181,182]. In a multicenter retrospective experience from Spain (SOTIS Study), ISA was used mainly as first-line therapy in 10 patients with post-transplant mucormycosis, with predominance of pulmonary involvement due to *Rhizopus* spp. The median treatment duration was 66.5 days, and six patients also received L-AmB as combination therapy. The clinical response rate at 6 weeks was 40.0%. Treatment-related adverse events occurred in one fifth of patients, although none of them required the premature discontinuation of ISA therapy [181].

Despite promising in vitro studies that suggest a synergistic activity against Mucorales for the combination of ISA and AmB [183,184], the clinical evidence supporting the use of antifungal combination therapy for IMI is particularly scarce [173]. In the previous mentioned study on post-transplant pulmonary mucormycosis, the 90-day all-cause mortality among patients that received combination therapy was 50.0% as compared to 16.7% in those treated with L-AmB alone, although the size of this former group was very small [72].

Regarding mechanisms of antifungal resistance in Mucorales, studies are scarce. It has been shown that the epimutation (RNAi-based silencing) of the genes targeted by tacrolimus and mammalian target of rapamycin (mTOR) inhibitors in *M. circinelloides* confers resistance to these agents for a short period [185]. These findings contrast with previous reports that suggest that calcineurin inhibitors are likely to enhance the activity of ISA against some Mucorales species but are in line with other studies suggesting the presence of antagonism or no effect between mTOR inhibitors (sirolimus) and broad-spectrum triazoles [186,187].

## 8. Prevention

There is no available experience with the use of triazole-based primary prophylaxis specifically directed against post-transplant mucormycosis [7]. In a single-center comparative study with 144 LuT recipients, the overall rate of breakthrough IFI with ISA prophylaxis was 7%, and no cases of mucormycosis were found (versus the rate of 8% with VORI, which included one case of mucormycosis) [182]. These figures align with the pooled incidence of proven or probable IFI (7%) reported in a meta-analysis of 991 patients (mainly with acute myelogenous leukemia or myelodysplastic syndrome and HSCT), with Mucorales accounting for 12% of the total [188]. Given the low absolute incidence of post-transplant mucormycosis and the ubiquitous nature of Mucorales, it is unlikely that further developments will provide conclusive evidence in this area.

## 9. Outcomes

The overall mortality of mucormycosis in SOT recipients is uniformly high, ranging from 45.2% to 48% across studies [5,21,59,61,72,73,175,176]. Outcomes have been found to be poorer for pulmonary and disseminated forms as compared to ROC or skin mucormycosis. Factors impacting 90-day mortality include disseminated infection, uncontrolled diabetes mellitus, and the number of immunosuppressive agents [16]. Unfortunately, recent studies point to the lack of significant improvements in the outcome of post-transplant pulmonary mucormycosis despite the advances made in the diagnostic approach [189].

## 10. Conclusions

Invasive mucormycosis represents a rare but life-threatening infection in the SOT population, characterized by a rapid progression and dismal prognosis. Despite advances in imaging techniques and molecular diagnostics and the introduction of novel antifungal agents such as ISA, clinical management remains challenging due to diagnostic delays and limited high-quality evidence guiding therapy. Early diagnosis, prompt initiation of effective antifungal treatment, surgical debridement when feasible, and a multidisciplinary approach remain the cornerstones of management. One of the limitations of our review is its narrative (rather than systematic) nature, which was restricted to studies in English language. Moreover, case reports from single centers may have been left out, thus limiting the scope of the manuscript. Future research should focus on improving early diagnostic tools, defining optimal therapeutic strategies (including the management of immunosuppression), and fostering multicenter collaborations to generate robust data to inform clinical practice.

## Figures and Tables

**Figure 1 jof-11-00853-f001:**
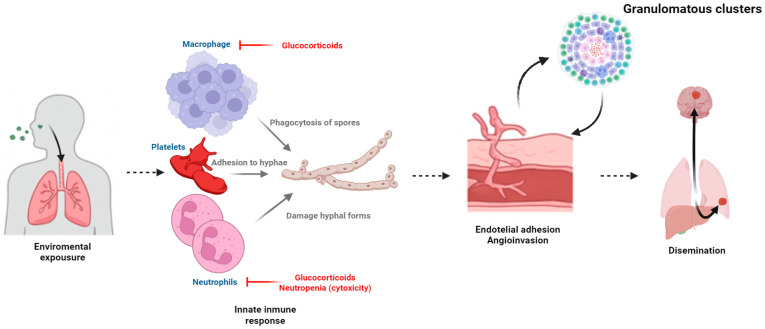
Pathogenesis of post-transplant mucormycosis (Created in BioRender. Fernández-ruiz, M. (2025) https://BioRender.com/refgacy).

**Figure 2 jof-11-00853-f002:**
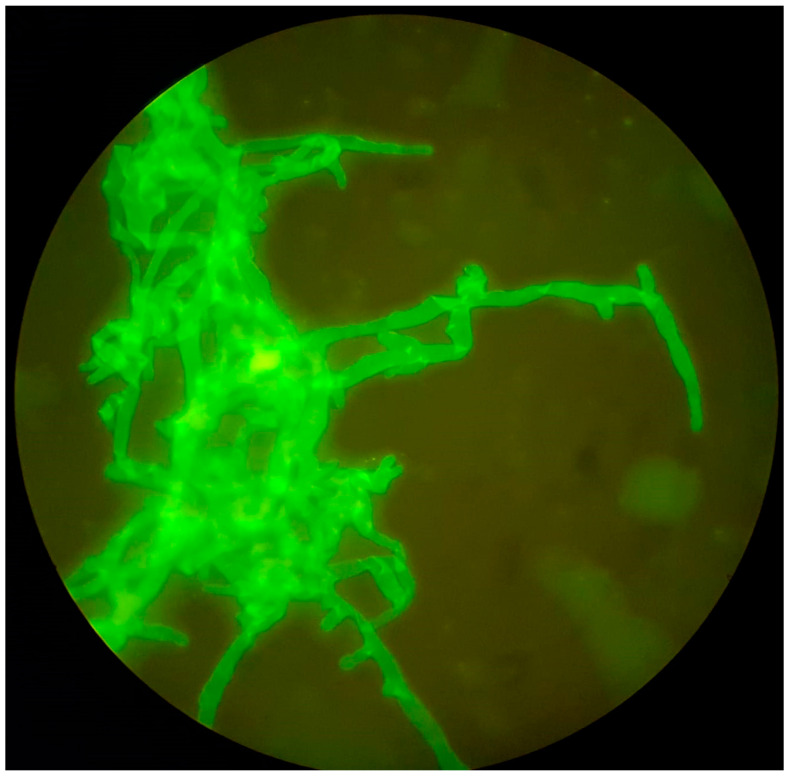
Broad, non-septate, ribbon-like hyphae consistent with Mucorales; Calcofluor-White staining ×200 (image provided by Dr. Ana Pérez de Ayala).

**Figure 3 jof-11-00853-f003:**
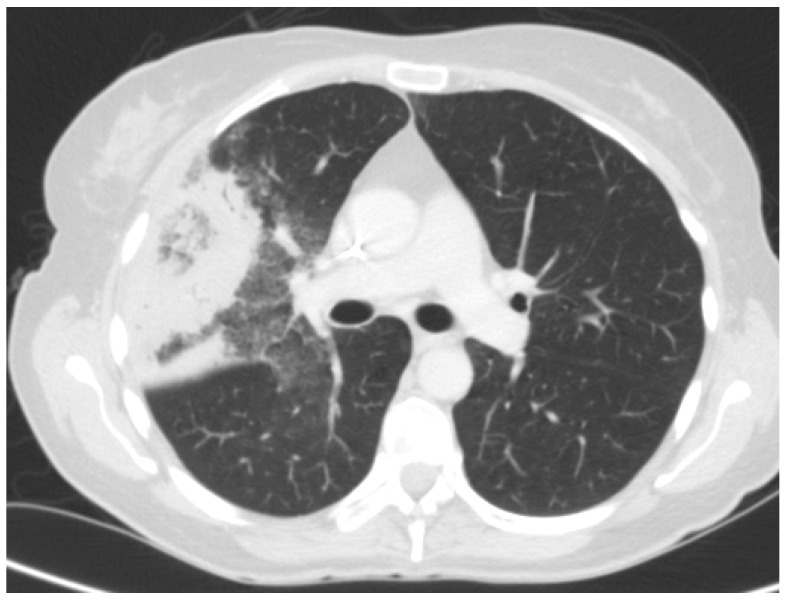
CT scan revealing a reversed halo sign in an HT recipient with pulmonary mucormycosis.

**Figure 4 jof-11-00853-f004:**
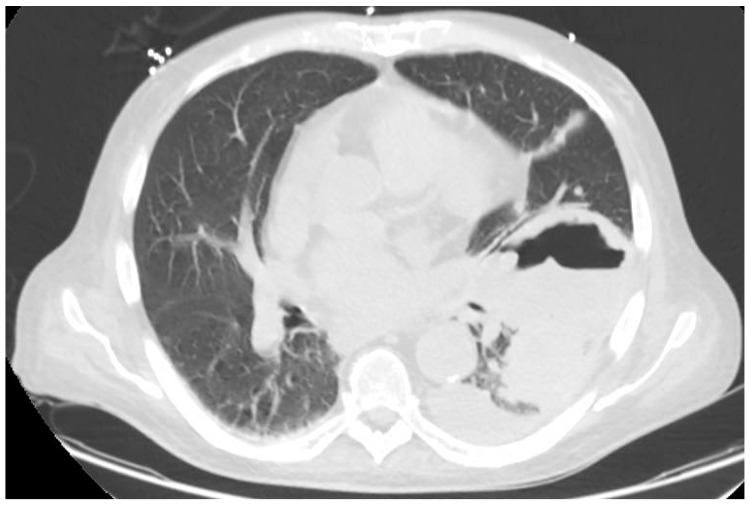
CT scan with the air crescent sign in an HT recipient with pulmonary mucormycosis.

**Figure 5 jof-11-00853-f005:**
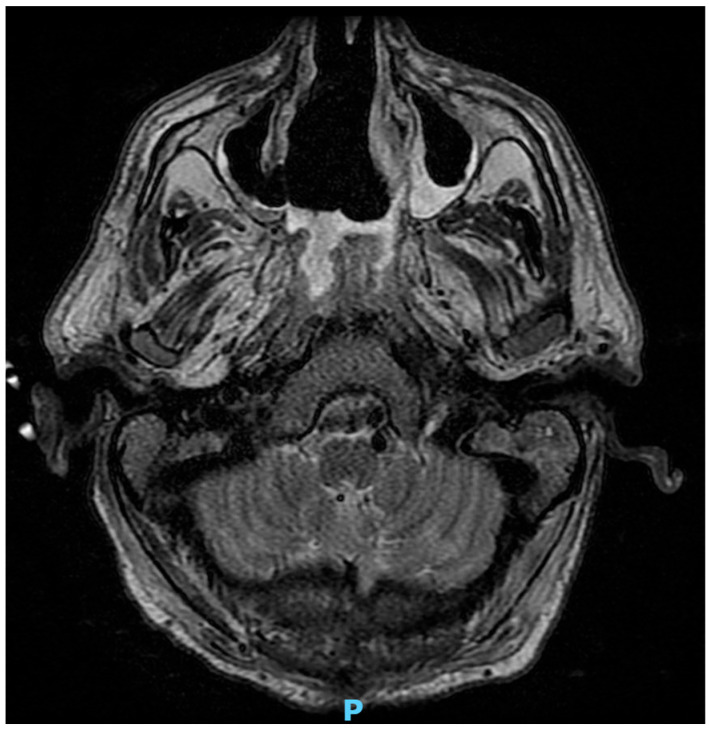
MRI with mucosal thickening of the left ethmoid sinus in a KT recipient with rhino-cerebral mucormycosis (P = posterior).

**Table 1 jof-11-00853-t001:** Most common causative agents of mucormycosis in SOT recipients in large cohorts and case series.

Mucorales Identified	Park, 2011 [12]	Palomba, 2024 [21]
	Nº of Cases (%)	Nº of Cases (%)
*Rhizopus* spp.	16 (57.1)	52 (41.9)
*Mucor* spp.	7 (25.0)	35 (28.2)
*Lichtheimia* spp.	0 (0.0)	20 (16.1)
*Cunninghamella* spp.	4 (14.3)	6 (4.8)
*Rhizomucor* spp.	0 (0.0)	5 (4.0)
*Apophysomyces elegans*	0 (0.0)	4 (3.2)
*Saksenaea* complex	0 (0.0)	1 (0.8)

**Table 2 jof-11-00853-t002:** Incidence of mucormycosis in multicenter cohorts and large single-center studies of SOT recipients.

Author, Year	Study Setting and Period	SOT Population	Nº of Cases (Incidence) of Post-Transplant Mucormycosis
Pappas, 2010 [18]	23 US centers (March 2001 to March 2006)	1063 SOT recipients (339 LT, 305 KT, 202 LuT, 91 HT)	28 (2.3%)
Neofytos, 2010 [15]	17 US centers (March 2004 to September 2007)	429 SOT recipients (108 LT, 106 LuT, 97 KT, 36 HT, 12 MVT)	10 (1.9%)
Hosseini-Moghaddam, 2020 [14]	Canadian administrative healthcare database (April 2002 to March 2016)	9326 SOT recipients (5685 KT, 1869 LT, 942 LuT, 453 HT, 333 KPT, 44 MVT)	11 (2.4%)
van Delden, 2020 [13]	7 Swiss centers (May 2008 to December 2014)	2761 SOT recipients (1612 KT, 577 LT, 286 LuT, 213 HT, 73 KPT)	4 (0.1%)
Permpalung, 2024 [19]	5 US centers (March 2020 to March 2022)	276 SOT recipients with COVID-19 (149 KT, 46 LT, 35 LuT, 28 HT)	1 (0.4%)
Gold, 2025 [17]	US health insurance claims dataset (January 2018 to December 2022)	9143 SOT recipients (5667 KT, 2025 LT, 759 HT, 650 LuT)	7 (0.4 per 1000 person-years)
Saxena, 2018 [16]	Single US center (January 2000 to December 2013)	584 pediatric SOT recipients (234 KT, 172 LT, 135 HT, 35 LuT)	1 (0.2%)
Fayos, 2025 [20]	Single Spanish center (January 2000 to December 2013)	712 KT recipients	1 (0.1%)

KPT: kidney–pancreas transplantation; KT: kidney transplantation; LT: liver transplantation; LuT: lung transplantation; MVT: multivisceral transplantation; SOT: solid organ transplantation.

## Data Availability

No new data were created or analyzed in this study. Data sharing is not applicable to this article.

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
