# Peer review of "The Unnoticed Threat: Clinical Characteristics, Risk Factors, and Outcome of Mucormycosis in Solid Organ Transplantation"

_jof, 2025, doi:10.3390/jof11120853_

Round 1
Reviewer 1 Report
In the present study, the authors present a narrative review of the current evidence on epidemiology, pathogenesis, risk factors, clinical presentation, diagnostic strategies, and treatment outcomes of mucormycosis in the solid organ transplant population. This is a well-presented and organised review providing all the spectrum of the infections saused by Mucorales, whose incidence has increased among solid organ transplant (SOT) recipients in recent years. The literature is updated and the conclusions of the study give new insights and prespectives for combating this pathogen. The English language is fine. Minor revisions are required before publication.
Major comment:
The authors have analysed the current knowledge on the antifungal therapy of mucormycosis and the molecular methods for diagnosis. Nonetheless, a description of the molecular Mechanisms of Antifungal Resistance in Mucormycosis is missing (please see at: Ganesan P, Ganapathy D, Sekaran S, Murthykumar K, Sundramoorthy AK, Pitchiah S, Shanmugam R. Molecular Mechanisms of Antifungal Resistance in Mucormycosis. Biomed Res Int. 2022 Oct 13;2022:6722245. doi: 10.1155/2022/6722245. PMID: 36277891; PMCID: PMC9584669.)
Author Response
Reviewer #1: Comments to the Author
Overall comment: In the present study, the authors present a narrative review of the current evidence on epidemiology, pathogenesis, risk factors, clinical presentation, diagnostic strategies, and treatment outcomes of mucormycosis in the solid organ transplant (SOT) population. This is a well-presented and organized review providing all the spectrum of the infections caused by Mucorales, whose incidence has increased among SOT recipients in recent years. The literature is updated and the conclusions of the study give new insights and perspectives for combating this pathogen. The English language is fine. Minor revisions are required before publication.
Response: We truly appreciate the positive feedback on our manuscript.
Detailed comment: The authors have analyzed the current knowledge on the antifungal therapy of mucormycosis and the molecular methods for diagnosis. Nonetheless, a description of the molecular Mechanisms of Antifungal Resistance in Mucormycosis is missing (please see at: Ganesan P, Ganapathy D, Sekaran S, Murthykumar K, Sundramoorthy AK, Pitchiah S, Shanmugam R. Molecular Mechanisms of Antifungal Resistance in Mucormycosis. Biomed Res Int. 2022 Oct 13;2022:6722245. doi: 10.1155/2022/6722245. PMID: 36277891).
Response: We thank the Reviewer for his/her constructive comments. We have included a brief description of the molecular mechanisms of resistance to antifungal agents exhibited by Mucorales (page 13 of the marked manuscript).
Reviewer 2 Report
The manuscript "The Unnoticed Threat: Clinical Characteristics, Risk Factors, and Outcome of Mucormycosis in Solid Organ Transplantation" presents updated information on the epidemiology, pathogenesis, risk factors, clinical presentation, diagnosis, treatment, and mortality of mucormycosis in solid organ transplant recipients. The main strength of the work is the relevance of the topic in the field of study, in addition to the appropriate structure and the clarity with which they present the information.
The weakness of this work lies in the fact that it is a narrative review, which only includes a search in a single database. There are no references in Spanish or any language other than English, which may introduce bias into the presented data. Therefore, my main suggestion is that you include a "Limitations" paragraph where you explain this situation.
It would be interesting to include a section on Prevention, given that the fungus is found in the environment. What preventive measures exist for transplant recipients?
Table 1. I believe it is advisable to include two columns: one showing the number of cases and another showing the proportion of cases, not grouped by genus, but rather showing the data separately for each species. If some Mucorales species have not been reported, such as Syncephalastrum racemosum, there is no need to include them in the table.
Figure 2. Indicate the objective lens used for observation and the staining technique employed.
Review and standardize the format of the references
Author Response
Reviewer #2: Comments to the Author
Overall comment: The manuscript "The Unnoticed Threat: Clinical Characteristics, Risk Factors, and Outcome of Mucormycosis in Solid Organ Transplantation" presents updated information on the epidemiology, pathogenesis, risk factors, clinical presentation, diagnosis, treatment, and mortality of mucormycosis in SOT recipients. The main strength of the work is the relevance of the topic in the field of study, in addition to the appropriate structure and the clarity with which they present the information.
Response: We truly appreciate the positive feedback on our manuscript.
Detailed comment: Table 1. I believe it is advisable to include two columns: one showing the number of cases and another showing the proportion of cases, not grouped by genus, but rather showing the data separately for each species. If some Mucorales species have not been reported, such as Syncephalastrum racemosum, there is no need to include them in the table.
Response: In response to this suggestion, we have modified Table 1 to depict in separate columns both the absolute and relative frequencies of causative agents of post-transplant mucormycosis identified in large case series (pages 1 and 2 of the marked manuscript).
Point 1. The weakness of this work lies in the fact that it is a narrative review, which only includes a search in a single database. There are no references in Spanish or any language other than English, which may introduce bias into the presented data. Therefore, my main suggestion is that you include a "Limitations" paragraph where you explain this situation.
Response: Indeed, one of the limitations of our work is that it is based on a narrative (rather than systematic) review of the literature, and case reports from single centers may have been left out, thus limiting its scope. Therefore, we have included a “Limitations” paragraph in the manuscript to highlight this point (page 14 of the marked manuscript).
Point 2. It would be interesting to include a section on Prevention, given that the fungus is found in the environment. What preventive measures exist for transplant recipients?
Response: The Reviewer has raised an interesting point. Unfortunately, there are very limited data on preventive measures specifically aimed at preventing post-transplant mucormycosis, despite the severity of this complication, likely due to the low absolute incidence in this population and the ubiquitous nature of Mucorales. We have included a new section to discuss this topic in the revised manuscript (page 13).
Point 3. Table 1. I believe it is advisable to include two columns: one showing the number of cases and another showing the proportion of cases, not grouped by genus, but rather showing the data separately for each species. If some Mucorales species have not been reported, such as Syncephalastrum racemosum, there is no need to include them in the table.
Response: Response: In response to this suggestion, we have modified Table 1 to depict in separate columns both the absolute and relative frequencies of causative agents of post-transplant mucormycosis identified in large case series (pages 1-2 of the marked manuscript).
Point 4. Figure 2. Indicate the objective lens used for observation and the staining technique employed.
Response: We have included the objective lens and the staining technique (page 13).
Point 5. Review and standardize the format of the references
Response: References have been reviewed and standardized as per the Journal style.
Reviewer 3 Report
A very well-prepared manuscript. The authors have done a tremendous job. In my opinion, the only thing missing is information about cases of ROC mucormycosis in immunocompetent patients. In my opinion, the only information missing is about cases of ROC mucormycosis in immunocompetent patients. At the same time, I would ask the authors to supplement their diagnostic approach with a few words about conventional diagnostic methods.
In my opinion, it would be worthwhile to make Figure 1 a bit more detailed or to add a second figure that would show the immune reaction during the development of mucormycosis in more detail.
Author Response
Reviewer #3:
Overall comment: A very well-prepared manuscript. The authors have done a tremendous job.
Response: We truly appreciate the positive comments by the Reviewer on our manuscript.
Detailed comment: In my opinion, it would be worthwhile to make Figure 1 a bit more detailed or to add a second figure that would show the immune reaction during the development of mucormycosis in more detail.
Response: We have accordingly modified the Figure 1. to better represent the host-pathogen interaction leading to the development of post-transplant mucormycosis (page 4 of the marked manuscript).
Point 1. In my opinion, the only thing missing is information about cases of rhino-orbital-cerebral (ROC) mucormycosis in immunocompetent patients.
Response: We appreciate this comment by the Reviewer. However, we would like to note that the scope of this review article was restricted to the SOT population. For the sake of clarity and conciseness, we have not included detailed descriptions of the different forms of mucormycosis in immunocompetent patients. On the other hand, the literature contains excellent reviews on this topic.
Point 2. At the same time, I would ask the authors to supplement their diagnostic approach with a few words about conventional diagnostic methods.
Response: We have accordingly included in the reviewed manuscript a brief section discussing conventional diagnostic methods (pages 10-11).
Point 3. In my opinion, it would be worthwhile to make Figure 1 a bit more detailed or to add a second figure that would show the immune reaction during the development of mucormycosis in more detail.
Response: We have accordingly modified the Figure 1. to better represent the host-pathogen interaction leading to the development of post-transplant mucormycosis (page 4).